# Sustaining Teaching with Technology after the Quarantine: Evidence from Chinese EFL Teachers' Technological, Pedagogical and Content Knowledge

**Fang Huang [1,*,†], Jiafu Qi [1,†]** and **Ailin Xie [1,2]**

1   School of Foreign Languages, Qingdao University, Qingdao 266071, China; jiafu@qdu.edu.cn (J.Q.);
    ailinxie0805@qdu.edu.cn (A.X.)
2   English Education Department, The Middle School Affiliated to Qingdao University, Qingdao 266073, China
*   Correspondence: huangfang@qdu.edu.cn
†   These authors contributed equally to this work.

**Abstract:** Given that little is known about English teachers' technological, pedagogical and content knowledge (TPACK), this study examined teachers' TPACK of using interactive whiteboards (IWBs) by contextualizing the research in the Chinese EFL context. Surveys and multi-case interviews were conducted among secondary school EFL teachers. The results revealed that Chinese EFL teachers generally perceived themselves to be competent in TPACK, with content knowledge achieving the highest value (5.545) and technological knowledge having the lowest value (5.147). In addition, teachers with higher professional titles perceived themselves as having lower TPACK. Barriers to using IWBs in English teaching include a lack of using efficacy regarding IWBs, traditional teaching beliefs, insufficient technical support and training, defects in IWBs for English teaching and time constraints. This study enriched technology adoption literature and informed policymakers and educational institutions of the necessity to provide specialized training to improve teachers' TPACK and take measures to reduce teachers' non-teaching-related tasks to ensure sustainable technology adoption in English teaching.

**Keywords:** EFL teachers; TPACK; sustain teaching; barriers; China

## 1. Introduction

Information and communication technology has brought great changes in people's study, work and life and has been promoting educational reform, improving teaching and learning effectiveness and, thus, sustaining quality education [1]. What roles technology plays and the degree to which it plays its roles in improving teaching and learning are largely dependent on teachers given that teachers are important agents in education and have volition in the decision-making process. Despite this, in many countries or regions, teachers' volitions were ineffective because of unexpected events, such as the COVID-19 pandemic that emerged in 2020 ([2]). For example, in the Chinese context, teachers at all educational levels were required to conduct online teaching to sustain students' learning without disruptions [2]. The success of remote teaching assumes that teachers are skilled to use online teaching tools and competent to deal with unexpected incidents during teaching. When schools were re-opened, teachers regained their voluntariness of using technology in teaching, but their perceptions of using technology may have changed, and their technological, pedagogical and content knowledge may also be different from the previous time (pre-COVID-19) when they could choose not to use technology. The extent to which teachers have sufficient TPACK determines whether they would be able to cater to students' learning needs and quality teaching. Therefore, teachers' TPACK and perceptions of technology-usage barriers in the post-COVID-19 period deserve revising among scholars.

In China, policies were put forward to promote EFL teachers to use technology (e.g., Internet-connected computers and interactive whiteboards) to simulate an authentic language-rich environment for students whose native language is Chinese [3,4]. The *Education Informatization 2.0* issued by the Ministry of Education in 2018 pointed out that teachers should transform teaching beliefs, update teaching methods, reshape their roles and improve their information literacy and competencies to cope with the fast changes that technology brings to education. What is more, *China's Education Modernization 2035* issued by the Ministry of Education in 2019 clearly stated that teachers should pay more attention to improving their competence in information literacy, profession and creativity. These national policies stressed an urgent need for teachers to improve their competence in using technology in teaching. Despite policy support and requirements, the problem of efficient technology use is still a major aspect of education as the COVID-19 pandemic uncovered real issues with that. Teachers' technology integration skills were tested, and the majority of countries and institutions have to admit to insufficient training and readiness to adopt innovative methods—where technology is not just an aspect or alternative but a real and the only effective tool to ensure the continuity of teaching and learning [2].

In the EFL teaching context, technology use enables teachers and students to access authentic English learning materials [2,5,6]. This is especially important for Chinese EFL teachers and students because English is not their mother tongue. Besides, technology use improves students' intercultural awareness and knowledge [2], language learning efficiency and effectiveness [7]. The usefulness of technology integration in English teaching has achieved consensus among EFL teachers, but the success assumes that EFL teachers have sufficient knowledge and skills to integrate technology in English teaching. To meet the demands of digital native students who are fond of using technology in learning [8], EFL teachers need to possess sufficient abilities to adopt technologies in teaching, and this requires teachers' perceptions of technology use, their beliefs about teaching [9,10] and technological, pedagogical and content knowledge (TPACK), as suggested by Mishra and Koehler [11].

As one of the innovative educational technologies, interactive whiteboards (IWBs) are extensively used in Chinese secondary schools. The functions that are default in IWBs include content displaying (texts, pictures, videos), digital writing and erasing and so on. These functions enable teacher–student interactions that reflect the student-centered pedagogy [12].

Although the government and administrations issued policies to promote technology integration in English teaching, EFL teachers' technological integration was not satisfactory [2,6]. Studies on EFL teachers' use of IWBs suggested teachers were either reluctant to use IWBs or used them at a low level. For example, previous studies suggested EFL teachers used IWBs to deliver content knowledge, but very few of them used them to communicate and interact with students [13,14]. Keeping such problems unaddressed will lead to a great waste of government investment, and the effectiveness of technology in English teaching will not be achieved.

Studies have been conducted to unpack barriers that teachers encountered when using technology. The literature review suggested that, besides the first-order barriers that were mainly related to external factors (time, hardware and software, support), teachers' beliefs about teaching and learning, self-efficacy and knowledge about using technology in teaching (i.e., TPACK) belong to the second-order barriers and play more important roles in influencing teachers' technology adoption [2,9]. However, studies about EFL teachers' perceptions and TPACK in the IWBs context are limited [15]. Previous studies about teachers' IWBs use in China are mostly descriptions of the definition, functions and teaching model [16], leaving EFL teachers' TPACK and perceived difficulties to remain unclear. Although there were studies on pre-service teachers' use of IWBs [17,18], research focusing on in-service teachers' IWBs usage is insufficient. Different from young pre-service teachers who grew up with more modern technology and are mostly technology-savvy, in-service teachers may have diverse attitudes regarding technology use. Nonetheless,

examining in-service teachers' perceptions is an urgent need to sustain teaching and learning with technology.

This study aims to investigate EFL teachers' TPACK and their perceptions of IWBs integration in English teaching. The research questions are as follows:

RQ1: How do EFL teachers perceive their TPACK when they use IWBs in teaching?

RQ2: What barriers do EFL teachers encounter when they use IWBs in teaching?

Unpacking these questions, this study will build on knowledge in both TPACK and technology acceptance theories. The research findings will also help governments to make informed decisions on investing in technology facilities and training as well as policymaking.

The following sections include a literature review of the existing studies on teachers' TPACK and uses of interactive whiteboards, the research methods, the results of the study, a discussion based on the research questions and a conclusion.

## 2. Literature Review

### 2.1. Technological, Pedagogical and Content Knowledge (TPACK)

Shulman [19] suggested that teachers should master both content knowledge and pedagogical knowledge, and the two kinds of knowledge interact with each other. This is what we know as pedagogical content knowledge (PCK). With the pervasiveness of technology use in education, Mishra and Koehler [11] have extended Shulman's framework by including technology as an important variable to measure teachers' knowledge and competence. They emphasized that teachers need to be able to integrate technology with specific teaching content in meaningful ways to enhance student learning. The framework of TPACK is gradually gaining support among teacher educators and evaluators.

According to Mishra and Koehler [11], the TPACK framework includes seven elements or categories: technology knowledge (TK); content knowledge; pedagogical knowledge; pedagogical content knowledge (PCK); technological content knowledge (TCK); technological pedagogical knowledge (TPK) and technological pedagogical content knowledge (TPCK). Besides, this framework also considers the dynamic and complex relationships among content, technology, pedagogy and context. Context refers to factors that exert influence on the overall TPACK development, such as prior knowledge, learning difficulties, etc. Figure 1 illustrates the TPACK framework.

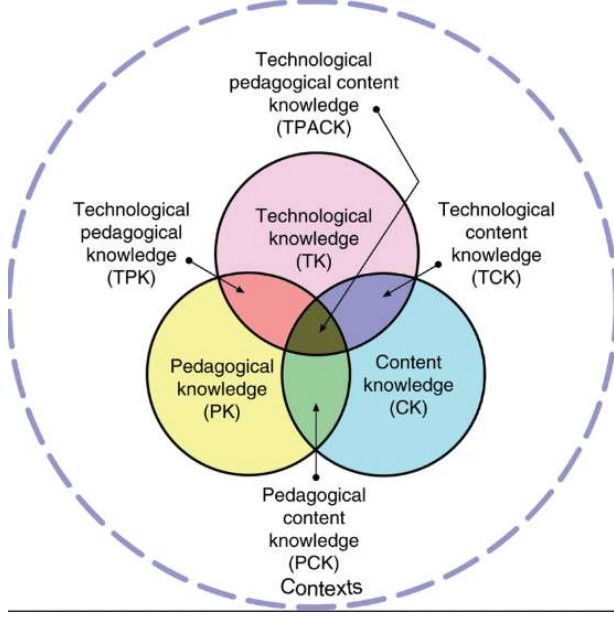

**Figure 1.** The framework of revised TPACK [11].

Research on TPACK is increasing recently [20]. To assess teachers' TPACK, both quantitative surveys and qualitative interviews were applied [21–24]. Researchers examined pre-service teachers' TPACK and provided suggestions to improve their TPACK [24–26]. Other studies examined the influence of TPACK on technology adoption (e.g., [27]). Studies also proposed that teachers' TPACK varies in teaching experience, gender, age and professional title [28,29]. In addition to these, teachers' beliefs about teaching influenced TPACK and actual technology use [30], while teachers' low level of TPACK and technology integration in teaching were attributed to factors such as insufficient teaching experience and technology training [31].

The literature review suggested that TPACK has become a useful framework to examine teachers' knowledge in teaching [32]. It is suitable and reasonable for this study to unpack EFL teachers' perceived barriers to use IWBs based on TPACK.

*2.2. Research on Interactive Whiteboards*

Interactive whiteboards (IWBs), as an innovative educational technology with various functions, are extensively used in classroom-based English teaching. Research on IWBs mainly focuses on two aspects, namely the benefits of IWBs in teaching and learning and factors affecting teachers' IWBs usage. Studies suggested using IWBs enables teachers to provide authentic learning materials and design interesting activities or games to meet students' needs [12,33]. Besides, IWBs result in enhanced presentations and the development of student motivation [16,34]. Compared to traditional lecture-based teaching, IWBs-based instruction improves students' learning achievements [35] and positively influences students' cognitive learning outcomes [12]. Therefore, most teachers hold positive attitudes toward IWBs usage in classroom teaching [36].

As for factors influencing teachers' adoption of IWBs, Tosuntaş et al. [37] suggested teachers' TPACK level significantly influenced their IWBs integration. Insufficient technological support, limited pedagogical knowledge and design thinking were reasons explained for teachers' low-level usage [2,16,21,38]. What is more, studies on pre-service teachers' IWBs use suggested that performance expectancy and social influence impact their intentions to use IWBs [39].

## 3. Methods

### 3.1. Participants

To answer the research questions, this study invited two groups of teachers who used IWBs in English teaching to participate in the survey and multi-case interviews.

Participants for the survey vary from 21 to 57 years in age (M = 37.9, SD = 0.883); their teaching experience ranged from 0 to 36 years (M = 14.7, SD = 1.938). Among them, 37 were males and 248 were females. As for the professional title, 116 teachers were awarded the primary title, accounting for 40.70%; 94 teachers had the intermediate title, accounting for 32.98% and 39 teachers had the senior title, accounting for 13.68%. In terms of teaching experience, 52.64% of them have less than 15 years of teaching experience. Further, 247 teachers reported that they had received training, accounting for 86.67%. They have average of 14.7 years' teaching experience (SD = 1.938). As for educational degrees, 2 teachers had junior college degrees, 249 teachers had bachelor's degrees and 34 teachers had master's degrees. Table 1 indicates the detailed information of participants.

In addition, semi-structured interviews with 6 teachers were conducted to understand barriers that teachers perceived when using IWBs in English teaching. Pseudonyms were used to represent interviewee's identity (see Table 2).

**Table 1.** Participants for quantitative study.

|  | Category | Frequency | Mean | SD |
|---|---|---|---|---|
| Gender | Male | 37 | / | |
| | Female | 248 | | |
| Age | ≤38 years old | 146 | 37.9 | 0.883 |
| | >38 years old | 139 | | |
| Professional title | Senior | 39 | / | |
| | Intermediate | 94 | | |
| | Primary | 116 | | |
| | Teacher to be decided title | 36 | | |
| Teaching experience | 0–5 years | 81 | 14.7 | 1.938 |
| | 6–15 years | 69 | | |
| | >15 years | 135 | | |
| IWBs Training | Yes | 247 | / | |
| | No | 38 | | |
| Educational degree | Junior college | 2 | / | |
| | Bachelor | 249 | | |
| | Master or higher | 34 | | |

**Table 2.** Participants for Interviews.

| Teacher | Gender | Age | Teaching Years | Professional Title | IWBs Using Experience | Interview Type |
|---|---|---|---|---|---|---|
| A | Female | 46 | 25 | Intermediate | 2 | Face-to-face |
| B | Female | 31 | 8 | Primary | 4 | Face-to-face |
| C | Female | 25 | 2 | No | 1 | Face-to-face |
| D | Female | 24 | 1 | Primary | 1 | Online |
| E | Male | 25 | 1 | Primary | 2 | Online |
| F | Female | 42 | 10 | Intermediate | 3 | Online |

*3.2. Instruments*

To examine EFL teachers' TPACK in the IWBs context, a questionnaire adapted from Archambault and Crippen [40] and Schmidt et al. [24] was used. It contains two parts. The first part includes questions about participants' demographic information, such as gender, age, teaching experience, professional title and educational degree. The second part consists of 37 items underlying the 7 constructs of TPACK. To be specific, it contains content knowledge (CK, 5 items), pedagogical knowledge (PK, 6 items), technology knowledge (TK, 5 items), pedagogical content knowledge (PCK, 7 items), technological content knowledge (TCK, 4 items), technological pedagogical knowledge (TPK, 5 items) and technological pedagogical content knowledge (TPCK, 5 items). All items were rated on a 7-point Likert scale (1 = strongly disagree, 7 = strongly agree). To ensure participants' understanding, two scholars who are experts in educational technology read through and checked items, and some wordings of items went through modification, deletion and addition to make them suit the research context. Items with ambiguous or overlapping meanings are revised to ensure face validity. For example, we have added examples for the item underlying TK, "I know how to use diverse functions of the IWBs"—(e.g., circle the content using digital pens default in the IWBs, design activities by using software default in IWBs)—"to help participants understand its meaning". The item underlying TCK dimension "I can adapt technology based on teaching needs" was deleted because its meaning is confusing and repetitive to other items in this dimension. Considering participants are Chinese teachers, survey items were presented in Chinese.

To deeply understand the barriers that teachers perceived when integrating IWBs in English teaching, semi-structured interviews were conducted based on an interview protocol. The authors changed the ways and sequence of asking where necessary.

### 3.3. Procedure and Data Analysis

Participants were informed of the research purposes and their right to withdraw from the study without a need to provide reasons. Online questionnaire was used to collect data for the quantitative section. Generally, participants spent about 10 min completing the survey. Descriptive statistics were used to analyze EFL teachers' perceptions of their TPACK levels. An independent sample *t*-test and one-way ANOVA were performed to examine whether teachers' TPACK varies in gender, age, educational degree, professional title and teaching experience.

The multi-case semi-structured interviews were conducted individually (face-to-face or online) to gain a deep understanding of EFL teachers' perceived barriers when they use IWBs. Each interview was conducted in Chinese to achieve better communication and understanding, and, on average, it lasted about 30 min. Additional questions were asked where necessary to achieve clarity or further understanding. All the interviews were completed in one month. Interviews were recorded based on agreement from interviewees and then transcribed word by word. The interview data analysis was conducted in a standardized way, through transcription, coding, data clustering, theme generation and conclusions.

## 4. Results

### 4.1. EFL Teachers' TPACK

The results of EFL teachers' perceptions of their TPACK were shown in Table 3. For the dimensions of TPACK, the mean values were all above 5, indicating that Chinese EFL teachers generally think that they have sufficient knowledge regarding the seven dimensions. The value of content knowledge (CK) achieves the highest score, while the value of technological knowledge (TK) is the lowest. These indicate that EFL teachers are very confident in their content knowledge (English language, such as linguistics and literature), but, comparatively, they do not feel very competent in technological knowledge regarding using IWBs. In addition, the value of TPK is also comparatively low, suggesting that teachers do not perceive that they have satisfying knowledge to combine technological knowledge and pedagogical knowledge. The mean values of CK, PK, TK, PCK, TCK, TPK and TPCK were 5.55, 5.50, 5.15. 5.54, 5.31, 5.20 and 5.26, respectively, and they were ranked as CK > PCK > PK > TCK > TPCK > TPK > TK.

**Table 3.** Descriptive results of EFL teachers' TPACK (N = 285).

| Constructs | Max | Min | Mean | SD |
|---|---|---|---|---|
| Content knowledge (CK) | 7 | 3 | 5.545 | 0.890 |
| Pedagogical knowledge (PK) | 7 | 3 | 5.499 | 0.813 |
| Technological knowledge (TK) | 7 | 3 | 5.147 | 0.949 |
| Pedagogical content knowledge (PCK) | 7 | 4 | 5.537 | 0.801 |
| Technological content knowledge (TCK) | 7 | 3 | 5.305 | 0.916 |
| Technological pedagogical knowledge (TPK) | 7 | 3 | 5.197 | 0.956 |
| Technological pedagogical content knowledge (TPCK) | 7 | 3 | 5.259 | 0.891 |

### 4.2. The Influence of Demographic Information on Teachers' TPACK

This study suggested EFL teachers' TPACK did not vary significantly by gender, age, teaching experience and educational degree, but there is a significant difference between teachers' TPACK and professional title. Besides, teachers with the primary title reach the highest TPACK level (5.436); teachers who were not awarded a professional title are mostly novice teachers and their TPACK ranked secondary to primary teachers (5.206).



Comparably, teachers with intermediate professional titles and those with senior titles have lower TPACK (5.168 and 5.000, respectively). The details are shown in Table 4.

**Table 4.** The influence of demographic information on teachers' TPACK (N = 285).

| Constructs | Dimensions | Number | Mean | Std. Deviation | Sig. (2–Tailed) |
|---|---|---|---|---|---|
| Gender | Male | 37 | 5.351 | 0.921 | 0.500 |
| | Female | 248 | 5.245 | 0.887 | |
| Age | ≤38 years old | 146 | 5.340 | 0.881 | 0.117 |
| | >38 years old | 139 | 5.174 | 0.896 | |
| Teaching experience | 0–5 years | 81 | 5.350 | 0.945 | 0.366 |
| | 6–15 years | 69 | 5.301 | 0.890 | |
| | >15 years | 135 | 5.182 | 0.857 | |
| Educational degree | Junior college | 2 | 6.000 | 1.414 | 0.084 |
| | Bachelor | 249 | 5.217 | 0.885 | |
| | Master or higher | 34 | 5.524 | 0.871 | |
| Professional title | Senior | 39 | 5.000 | 0.747 | 0.028 ** |
| | Intermediate | 94 | 5.168 | 0.847 | |
| | Primary | 116 | 5.436 | 0.918 | |
| | Not awarded | 36 | 5.206 | 0.976 | |

Note: ** $p < 0.01$.

### 4.3. Perceived Barriers to IWBs Integration

EFL teachers perceived some barriers when they used IWBs, including a lack of IWBs-using efficacy, traditional teaching beliefs, *insufficient technical support and training*, defects of IWBs for English teaching and time constraints. The details are elaborated in the following sections.

#### 4.3.1. Lack of IWBs-using Efficacy

EFL teachers generally think that their IWBs usage is not satisfying and needs to be further improved, especially for elderly teachers. This is aligned with previous studies among university EFL teachers in China (e.g., Ref. [2]). Designing activities using IWBs to promote interaction is critical to improve students' learning motivation, engagement and learning outcomes [41], but teachers stated that, sometimes, they are not able to use the IWBs flexibly and creatively.

> . . . *Sometimes I cannot control the time appropriately when I use IWBs in English teaching . . . I don't know how to use the IWBs to design communicative activities, but I know I should improve my competence in using IWBs . . .* (Teacher C)

> . . . *Compared with young teachers, I think my ability to IWBs use is relatively low. Some young teachers learn things very fast; I have been learning how to use it, but it is very hard, and till now I still feel not good because of my limited ability . . .* (Teacher A)

#### 4.3.2. Traditional Teaching Belief

Influenced by traditional teaching beliefs [4], Chinese teachers usually deliver content to students and require them to memorize content. Teaching and learning are, to a large degree, exam-oriented, and, thus, students are accustomed to cramming knowledge and rote memory is regarded as an effective strategy to achieve good scores. EFL teachers usually pay more attention to scores in the summative assessment than to students' learning growth reflected in formative assessment. Guided by the traditional teaching belief, some teachers perceive IWBs use may distract students' attention and is useless for students' learning; thus, they are unwilling to explore the functions of IWBs in English teaching.

> . . . *Our teaching is mostly test-oriented, and we pay more attention to students' scores. If I use the IWBs during the whole class, I might have to repeat the language points in the next class . . .* (Teacher A)

> *. . . I think IWBs sometimes distracted students' attention in class, and students still need to pay attention to linguistic points in the end . . .* (Teacher E)

> *. . . Teachers speak most of the time in class, and students just listen to what teachers say and take notes. There is limited time for students to interact with the computer. Although there are various functions in the IWBs, we still choose to spend more time explaining linguistic points . . .* (Teacher B)

### 4.3.3. Insufficient Technical Support and Training

EFL teachers perceived IWBs as an innovative educational technology. They did not receive enough training, and, therefore, IWBs are difficult for them to use, especially for elder teachers. Many teachers complained that technical support in their schools is very limited, which affected their willingness to adopt IWBs in English teaching. In addition, although some schools provided teachers with technology training, their training lacked integration with the teaching content (English), so, still, English teachers felt using IWBs in English teaching was challenging.

> *. . . We have not been trained to use the IWBs . . . There was no technical support for us when technical problems occurred in class. We need to cope with difficulties by ourselves, but it was very time consuming . . .* (Teacher C)

> *. . . I once encountered a problem with the whiteboard in class, and I tried my best to fix it for a long time, but it was in vain. If we could call someone for help or seek for technological support, it would have been handled quickly . . .* (Teacher A)

> *. . . We got technological training in our schools to teach teachers how to use the basic functions of IWBs, but in fact, we still don't know how to use it to teach English . . .* (Teacher F)

### 4.3.4. Defects of IWBs for English Teaching

Although English teachers learn English for years, teachers suggest Mandarin is still the dominant instructional language in English class. Given that English speaking is one of the required exam sections in the senior high school entrance examination, English teachers seek ways to improve students' oral English. However, EFL teachers suggest the materials default in IWBs are not related to English textbooks and are not helpful to English-speaking tests. When they designed the teaching plans, they could not find useful English-speaking materials from the IWBs. Besides, elderly teachers suggest it is difficult to design English teaching by using IWBs.

> *. . . My English pronunciation is not very standard, so I hope that students can read after good materials, for example, recordings from native speakers . . . it would be better if IWBs stores textbook reading recordings so that teachers can use them . . .* (Teacher E)

> *. . . I hope the IWBs could have the dictation function by using recordings from native speakers because the high school entrance exam pays attention to students' listening comprehension . . .* (Teacher D)

> *I hope it can provide materials about economy we need from the IWBs resource store or provide a similar teaching example closely related to the textbook, which may be much better for elder teachers.* (Teacher F)

### 4.3.5. Time Constraints

Secondary EFL teachers suggested that they have a heavy workload and, thus, they barely had sufficient time to design teaching, especially in terms of technology integration in teaching steps. They also suggested they must deal with many non-teaching-related tasks, which influenced the time and effort they would spend on teaching design. In addition, the limited English teaching hours make them feel designing interactive activities involving technology use is an impractical operation. Thus, despite that EFL teachers are generally aware of the importance of integrating technology in English teaching, they do

not have enough time to think about innovative and creative technology use. Unless they were given an opportunity to demonstrate teaching in an open class, they did not use IWBs at a satisfactory level.

> . . . *I think junior high school English teaching and learning is under great pressure because students need to remember many language points within a limited class time. It is also hard for teachers to finish teaching tasks in a traditional way, let alone using the IWBs to do interactive activities, which waste class time* . . . (Teacher B)

> . . . *Due to the limited class time, it is a challenge for us to come up with the teaching schedule, so I use it less in my daily teaching. But I will use it in the open classes* . . . (Teacher C)

> . . . *We have to deal with many tasks unrelated to English teaching especially COVID-19 to sustain formal teaching and learning* . . . *These tasks affect our teaching schedules, and we have no time to design teaching to integrate IWBs although we know we should do so* . . . (Teacher F)

## 5. Discussion

### 5.1. EFL Teachers' TPACK

From the results of EFL teachers' perceived TPACK (Table 3), we can understand that Chinese EFL teachers perceived themselves as having obtained sufficient content knowledge, while technological knowledge of using the IWBs was comparatively lower. This echoes the findings from previous studies (e.g., [42]). In addition, EFL teachers' knowledge related to technology (technological knowledge, technological and pedagogical knowledge, technological, pedagogical and content knowledge) was insufficient compared to those without involving technological knowledge (content knowledge, pedagogical knowledge), aligned with previous studies (e.g., [43]). The possible reasons were that, for most EFL teachers, the interactive whiteboards (IWBs) are innovative and they were not very familiar with the functions installed in them. In addition, the participants in this study were mostly experienced teachers who did not grow up with technology, and, thus, they may lack technology-using experience. Moreover, the EFL teachers in this study did not receive sufficient teacher training to learn to integrate IWBs in English teaching. Although Ertmer et al. (2012) [9] suggested the first-order barriers (e.g., time, training, facilities) for teachers' technology adoption play peripheral roles, they are still prominent in the Chinese EFL context, especially for experienced teachers who use innovative technologies.

The results of the study suggested that teachers with senior professional titles have significantly lower TPACK compared with their counterparts. On the one hand, this may be attributed to the fact that teachers with senior professional titles might gradually lose their motivation to sustain professional development (e.g., [44]). Wang and Wu [44] indicated that, after earning the senior title, teachers' motivation for teaching waned, and many of them were promoted to be administrative leaders, making adopting innovative teaching skills unnecessary to survive. For those with lower-level professional titles, they strive to obtain knowledge and improve teaching competence because of its importance in teaching evaluation and promotion [45]. On the other hand, teachers with senior professional titles may have more opportunities to be exposed to excellent teaching and, therefore, they might perceive their current knowledge status as unsatisfying. Therefore, their perceived TPACK might be low. However, this situation needs to be further explored to achieve a better understanding.

### 5.2. EFL Teachers' Barriers when Using IWBs

In line with the previous findings [46], this study suggested that EFL teachers perceived a lack of technology efficacy, especially for elder teachers. It is understandable given that the IWBs are new educational technology for EFL teachers. Teachers' self-efficacy with technology plays an important role in affecting their attitudes and ways they integrate technologies into teaching [47]. If teachers possess higher self-efficacy for technology

integration, they would be likely to have higher TPACK and be more confident in using technology [48]. Therefore, EFL teachers should try their best to improve their IWBs competence, supported by the government and school leaders.

As the second-order barriers indicating an individual user's underlying beliefs about teaching and learning [9], Chinese EFL teachers' pedagogical beliefs are crucial to their technology adoption [2]. In the Chinese teaching context, the high school entrance examination and college entrance examination are regarded as the most fundamental and crucial exams for both students and teachers, and, thus, teachers focus more on whether the use of IWBs can improve students' test scores. Although using technology may improve students' interests [35], it might not help students to significantly improve their test scores within a short time. Be that as it may, teachers' teaching beliefs need to be reframed to improve students' overall development, and technology use enables students to improve their thinking and communication abilities [49].

The technological training that teachers received was not tailored to English teachers' needs [50]. The existing technological training only teaches some basic functions of IWBs, which contributed limitedly to improve teachers' teaching ability. Only when teachers know how to integrate IWBs with English teaching content and pedagogy would they be able to integrate technology in a good way [51]. Technology training related to English teaching is exactly what EFL teachers are eager for. What is more, it is understandable that technical support is insufficient in secondary schools given that Chinese schools lack teachers, not to mention teacher support staff.

Time constraints deeply influenced the whole teaching design and procedure, including the use of technology in classroom teaching [9,52]. EFL teachers in this study have heavy workloads, especially during COVID-19 and in the post-COVID-19 period. In the Chinese educational context, teachers are fully responsible for students' matters in schools. They were not only responsible to deliver lessons, but also deal with other things, such as students' security and conflicts and so on. It is not helpful to sustain professional development if they are occupied with too many non-teaching-related tasks.

### 5.3. Limitations and Suggestions for Further Study

The current study is limited in sample size considering the large population and the geographical diversity in China. Due to the impact of the COVID-19 pandemic, this study adopted an online questionnaire to collect interview data, which is not the best way to generate deep understanding [53]. In addition, this study did not report students' perspectives and their roles in influencing teachers' technology adoption. Therefore, further studies are suggested to expand the sample size, conduct face-to-face interviews and widen research considerations (e.g., students' roles) to gain a deeper understanding of EFL teachers' use of IWBs and their barriers when they integrate them in English teaching.

### 6. Conclusions

This study explored Chinese secondary EFL teachers' perceptions of their IWBs integration based on the TPACK framework and unpacked barriers that influence EFL teachers' IWBs integration. The results of this study contribute to people's understanding of EFL teachers' technology uptake to sustain teaching and learning in the post-COVID-19 period. This study provided empirical evidence for policymakers to organize teacher training programs by considering English teaching content and pedagogy. It is also important for governments to reform language assessment so that teachers' teaching beliefs may be reconstructed and, when they teach, they may better serve students' overall development. Besides, technological training and support are urgently needed in secondary schools to reduce teachers' technology-using anxiety. Last but not least, policymakers and educational administrators are suggested to take measures to reduce non-teaching-related tasks to enable teachers to devote themselves to teaching.

**Author Contributions:** Conceptualization: F.H.; Methodology: F.H. and A.X.; Writing—Review & Editing: F.H. and A.X.; Formal Analysis: A.X.; Supervision: F.H.; Project Administration: F.H. and J.Q.; Funding Acquisition: J.Q. All authors have read and agreed to the published version of the manuscript.

**Funding:** This study was supported by the project titled "EFL teachers' perceived information literacy and influencing factors in technology-enhanced teaching" (21WYJYZD10) and the project titled "National New liberal Arts Studies and Reform Implementation: Cultivating German Talents Creativity" (2021110061).

**Institutional Review Board Statement:** The study was conducted in accordance with the Declaration of Helsinki and was approved by the Academic Committee of School of Foreign Languages at Qingdao University (Approval code 20210630WY and date of approval 30 June 2021).

**Informed Consent Statement:** Informed consent was obtained from all study participants at the time of initial data collection.

**Data Availability Statement:** Data supporting reported results are available on request.

**Acknowledgments:** The authors appreciate all participants in this study.

**Conflicts of Interest:** The authors declare no conflict of interest.

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
