# Peer review of "Sustaining Teaching with Technology after the Quarantine: Evidence from Chinese EFL Teachers’ Technological, Pedagogical and Content Knowledge"

_sustainability, doi:10.3390/su14148774_

Round 1

Reviewer 1 Report

Post pandemic context and use of IWBs (technology) needs to be interconnected more closely, as when writing proceeds, the context has faded. Regarding the quantitative research part, the researcher could elaborate more substantially how data collection process is validated and closely scrutinized, e.g. wordings modification, overlapping meanings, etc. Discussions could be more in-depth, with reference to theoretical basis to explain the perceived and/or actual difficulties of using technology with reference to the factors of authenticity, specifications and relevance on teachers. Yet, how is the role of students in aggravating the difficulties? The other side of the coin should also be considered. In the discussion section, alignment with previous findings has been mentioned several times, this gives the impression that the research topic has no added value on scholarship. Please enhance the language proficiency for accuracy and logical fluency of the content. 

Author Response

Reviewer 1:

Post pandemic context and use of IWBs (technology) need to be interconnected more closely, as when writing proceeds, the context has faded.

Response: Thank you for your suggestion. We have added some content to connect the context and the use of IWBs in text. Please see page 2: “Despite policy support and requirements, the problem of efficient technology use is still a major aspect of education as the COVID-19 pandemic uncovered real issues with that. Teachers’ technology integration skills were tested, and the majority of countries and institutions have to admit the insufficient training and readiness to adopt innovative ways – where technology is not just an aspect or alternative, but a real and the only effective tool to ensure the continuity of teaching and learning [2]”.

Regarding the quantitative research part, the researcher could elaborate more substantially how data collection process is validated and closely scrutinized, e.g., wordings modification, overlapping meanings, etc.

Response: Thank you. We have added more details to explain the process in the text. Please see pages 5-6. 

Discussions could be more in-depth, with reference to theoretical basis to explain the perceived and/or actual difficulties of using technology with reference to the factors of authenticity, specifications and relevance on teachers. Yet, how is the role of students in aggravating the difficulties? The other side of the coin should also be considered.

Response: Thank you for your suggestion. We have elaborated the discussions in text to refer to theoretical basis (page 10). For example, we have added, “Although Ertmer et al (2012) [9] suggested the first-order barriers (e.g., time, training, facilities) for teachers’ technology adoption play peripheral roles, they are still prominent in the Chinese EFL context, especially for experienced teachers who use innovative technologies”. Also, we definitely agree with your idea to consider students’ roles, but the focus is on the teachers’ side, and we do not have data from students’ perspectives. So, we have described this in the limitation and suggested it as a future study direction. Please refer to the page 11.

In the discussion section, alignment with previous findings has been mentioned several times, this gives the impression that the research topic has no added value on scholarship. Please enhance the language proficiency for accuracy and logical fluency of the content.

Response: Thank you for your suggestion. We have revised the language to make it accurate, logical, and fluent. For example, on page 9, we have used “echoes the findings from the previous study”. 

Reviewer 2 Report

The main idea is very good, objectives are stated clearly, the course of study is clear and easy to follow. The authors are trying to tackle the issues of efficient use of technology by specific category of teachers – English teachers. That has allowed to track the main problems in this specific category so that they can be managed in an informed and efficient way.

 The problem of efficient use of technology today is a major aspect of education as COVID uncovered real issues with that. “Ticking the box” approach was challenged by pandemic where skills in using technology were tested and majority of countries have to admit the insufficient training and readiness to adopt new ways – where technology is not just an aspect but a real and the only tool to deliver the education.

 The literature review is solid and proves the gap in research on the matter as well as the urgent need to address the problem.

 Th methodology of the experiment is clearly presented and the methods are appropriate.  The idea to combine the questionnaire and the interview adds the depth and reliability  to the findings.

Results are clearly presented in RESULTS and DISCUSSION sections. 

 Some minor grammar and style issues have to be addressed, some examples are below:

 they have can choose not to use technology.

 the research questions are named RO (objectives i suppose)

The findings in 4.2 suggest there is no proper link to age - however there is link to position - higher ranking is linked to lower competence - could it be linked to age as well- as higher title presupposes older age -please check, but that is just suggestion

Author Response

Reviewer 2:The main idea is very good, objectives are stated clearly, the course of study is clear and easy to follow. The authors are trying to tackle the issues of efficient use of technology by specific category of teachers – English teachers. That has allowed to track the main problems in this specific category so that they can be managed in an informed and efficient way. The problem of efficient use of technology today is a major aspect of education as COVID uncovered real issues with that. “Ticking the box” approach was challenged by pandemic where skills in using technology were tested and majority of countries have to admit the insufficient training and readiness to adopt new ways – where technology is not just an aspect but a real and the only tool to deliver the education. The literature review is solid and proves the gap in research on the matter as well as the urgent need to address the problem. Th methodology of the experiment is clearly presented, and the methods are appropriate.  The idea to combine the questionnaire and the interview adds the depth and reliability to the findings. Results are clearly presented in RESULTS and DISCUSSION sections.

Response: Thank you for your support of our manuscript.

 Some minor grammar and style issues have to be addressed; some examples are below:

they have can choose not to use technology.

Response: Thank you, we have revised it in text.

 the research questions are named RO (objectives i suppose)

Response: Thank you. We have corrected the typo in text.

The findings in 4.2 suggest there is no proper link to age - however there is link to position - higher ranking is linked to lower competence - could it be linked to age as well- as higher title presupposes older age -please check, but that is just suggestion

Response: Thank you. We have double-checked the data and no mistake was found in this section. We believe possible reasons are that teachers with lower professional ranking may not necessarily be young, some of them might be middle-aged and elder teachers and they remain unpromoted.